# One-pot synthesis of cyclic-aminotropiminium carboxylate derivatives with DNA binding and anticancer properties

Bibhuti Bhusana Palai[1,2,4], Saket Awadhesbhai Patel[2,3,4], Nagendra K. Sharma[1,2 ✉] & Manjusha Dixit [2,3 ✉]

Tropolone, a nonbenzenoid aromatic molecule, is a constituent of troponoid natural products possessing a wide range of bioactivities, including anticancer. This report describes the one-pot synthesis and mechanistic studies of fifteen fluorescent $C^{aryl}$-$N^{alkyl}$-substituted cyclic-aminotroponiminium carboxylate (cATC) derivatives by unusual cycloaddition and rearrangement reactions. Herein, the biochemical studies of four cATC derivatives reveal a non-intercalative binding affinity with DNA duplex. In vitro/in vivo studies show strong anti-tumor activity in three cATC derivatives. These derivatives enter the cells and localize to the nucleus and cytoplasm, which are easily traceable due to their inherent fluorescence properties. These three cATC derivatives reduce the proliferation and migration of HeLa cells more than the non-cancer cell line. They induce p38-p53-mediated apoptosis and inhibit EMT. In xenograft-based mouse models, these cATC derivatives reduce tumor size. Overall, this study reports the synthesis of DNA binding fluorescent $C^{aryl}$-$N^{alkyl}$-cyclic-amino-troponiminium derivatives which show anti-tumor activity with the minimum side effect.

[1] School of Chemical Sciences, National Institute of Science Education and Research (NISER) Bhubaneswar, PO: Jatni 752050 Bhubaneswar, Odisha, India. [2] Homi Bhabha National Institute, Training School Complex, Anushaktinagar, Mumbai 400094, India. [3] School of Biological Sciences, National Institute of Science Education and Research (NISER) Bhubaneswar, PO: Jatni 752050 Bhubaneswar, Odisha, India. [4]These authors contributed equally: Bibhuti Bhusana Palai, Saket Awadhesbhai Patel. ✉email: nagendra@niser.ac.in; manjusha@niser.ac.in

Tropolone, non-benzenoid aromatic molecule, is a major constituent of *Troponoid* natural products, which are mostly found in plants and fungi[1–3]. Troponoids have wide-range of bioactivities such as *antibacterial, antifungal, antiviral, insecticidal, anticancer*[4–6]. Their bioactivities mainly depend upon the metal-complexation with metalloenzymes[7]. The synthetic analogs of tropolones such as *aryl-tropolones, amino-tropones*, and *aminotroponimines* also exhibit similar types of metal complexing properties[8–10]. Synthetic derivatives, *β-phenyl-tropolone*, and aryl-2-aminotropone have shown remarkable bioactivities including anticancer[11–13]. The tropolone scaffold provides ample sites for the chemical modifications which can facilitate the development of the molecules with stronger anticancer activity and specificity. Another category of synthetic derivatives, 2-aminotropones and aminotroponimines, exhibit better metal-chelating properties as compared to tropolone owing to the amine functionality[14,15]. The cyclic derivatives of amino-troponimines and troponoid-pyrazine derivatives are poorly explored as compared to tropolone. However, benzenoid pyrazine derivatives comprise useful bioactivities including DNA duplex binding through intercalative mode[16–20]. The non-benzenoid benzopyrazine derivative, 2,3-dihydro-1*H*-cyclohepta[b]pyrazine are yet to be explored in the development of anticancer agents (Fig. 1A)[21]. Tropolone derivatives also exhibit unique photophysical properties, including fluorescence, mainly owing to π-π*, n-π*, charge transfer, and the delocalization of *π-electrons* through the formation of *troponium-cation* at tropone ring system[22–27]. However, their fluorescence quantum yield is very low in polar solvents. The *N*-substituted *cyclic*-aminotroponimine derivatives could form stable *aminotroponiminium-cation* entities, comprising better delocalization of conjugated π-electrons and cationic charge separation at tropone-ring that could lower electronic transition energy and improve fluorescence properties. It would be lucrative to explore the structural and functional properties of substituted *cyclic*-aminotroponimine derivatives. We rationally designed the substituted *cyclic*-aminotroponimines derivatives from previously reported *troponyl-ketene* intermediate of unnatural *troponyl*-amino acid and *aryl-alkyl imine* via [2 + 2] cycloaddition reaction (Fig. 1B)[28–30]. This report describes the one-pot synthesis of novel *N*^*Alkyl*/*C*^*Aryl*-substituted *cyclic*-amino-tropiminium carboxylate (cATC) derivatives from *N*-alkyl-troponyl glycinate ester and *aryl-alkyl imine* derivatives by cycloaddition, and a unique rearrangement under mild acidic anhydrous conditions. This article also describes their structural

analyses, photophysical properties, and DNA binding studies. Further, we have investigated cATC derivative's anticancer properties. So far, there is no report about substituted *cyclic*-aminotropimine derivatives. Hence, our results have huge scope for developing the tropolone-based synthetic fluorescent anticancer drug candidates, which could be traceable in the cellular environment at real time.

## Results

**Synthesis.** We began the synthesis from commercially available Tropolone (1) molecule by following the synthetic route depicted in Fig. 2. Tropolone (1) was modified into the unnatural amino acid derivative (2), *N*-troponyl-*N*-Phenylethyl glycinate ester by following our previously reported procedure[29]. Their characterization data (NMR/HRMS/FT-IR) are provided in the Supplemental Material (SM). Their spectra (NMR/HRMS) are depicted Supplementary Data 1 (Fig. S1–S34). Pleasantly, we obtained the single crystal of troponyl-amino acid ester (2) in the organic solvent system MeOH:DCM (1:20) and studied by X-ray diffractometer. Its structural details (ORTEP diagram and X-ray parameters) are provided in the Supplemental Material (Fig. S1 and Table S1) and Supplementary Data 2. Crystal data has been submitted to Cambridge Crystallographic Data Centre (CCDC) with reference number 2035166, which confirmed the structure of *tr-pheneg* ester (2). Next, we synthesized various *imines* derivatives (5) from aryl *aldehydes* (3a-3i) and aliphatic *amines* (4a-4f) at the expense of water (H_2O) elimination by following the reported procedure (Fig. 2)[31]. In situ, we generated reactive *troponyl-ketene* intermediate (2*) from troponyl glycinate ester (2) under mild acidic conditions (5.0% TFA in anhydrous CH_3CN) and treated with excess *arly-alkyl imine* (5a) that was prepared from benzaldehyde (3a) and benzylamine (4a). After completion of reaction, we purified the new product (6a) through the *silica-gel* column and characterized it by NMR/ESI-HRMS/single-crystal x-ray techniques. Their analytical data are provided in the Supplemental Material. The NMR and mass data of isolated products confirm the formation of an adduct containing *troponyl-ketene* (2*) and imine (5a) residue. ¹H-NMR of 6a supports the presence of troponyl ring and aryl ring protons in the aromatic regions (deshielded) along with glycinate/alkyl amine protons in aliphatic regions (shielded). The ¹³C-NMR of product 6a exhibits two new peaks (~δ150.0 ppm and ~δ152.0 ppm) along with glycinate carboxylate carbonyl (–C = O) at ~δ170 ppm (Table S2, Entry 1) at the expense of characteristic troponyl carbonyl (*tr*-C = O) peak of reactant 2 at

**(A) Previous Report**

Tropolone     Aminotropone     Aminotroponimine     2,3-dihydro-1*H*-cyclohepta[b]pyrazine

**(B) This Report:** *Non-benzenoid pyrazine derivatives*

*Cyclic*-aminotroponiminium carboxylate (cATC)          *Ketene*

*N,N*-dialkyl-dihydrocyclohepta[b]pyrazinium carboxylate

**Fig. 1 Chemical structures of Tropolone derivatives. A** Previously reported Troponyl derivatives. **B** Rationally designed cATC derivatives.

~δ182.0 ppm along with other carbon peaks of aryl/aliphatic residues. To ensure the structure of product **6a**, we crystalized the isolated product in organic solvent systems. We also obtained the

single crystal of product (**6a**) in the organic solvent system MeOH:DCM (1:20). Its structural details (ORTEP diagram and X-ray parameters) are provided in the Supplemental Material

**Fig. 2 Chemical synthesis of cATC derivatives from tropolone and imines. A** Formation of Ketene intermediate that reacts with imine and form cATC derivative. **B** Structure of various cATC derivative from different imines.

**Fig. 3 The plausible reaction mechanism for the formation of cATC derivatives.** The formation of cATC derivative by a [2 + 2] cycloaddition between *ketene* and *imine* followed by unique rearrangement.

(Fig. S2 and Table S3) and Supplementary Data 3. The single crystal of cATC derivative **6a** is hydrogen bonded with $H_2O$ molecules as –O = C–O----H–O–H at distance 1.9 Å (SM, Fig. S2). Its X-ray analysis confirms the structure of the product as the substituted cATC derivative (**6a**), *N-benzyl-N-phenylethyl phenyl troponyl dihydropyrazinium carboxylate*. The crystal data has been submitted to CCDC with reference number 1889119. We repeated the reaction with *tr-pheneg* glycinate ester (**2**) and different imines (**5**) under similar conditions and characterized the respective products (**6b-6o**). Similarly, we purified and characterized products (**6b-6o**). Their characterization data along with comparative $^{13}$C-NMR are provided in the Supplemental Material. Their NMR and mass analysis results are consistent and support the formation of cATC derivatives. We also obtained the single crystal of cATC derivatives (**6d/6k**). Their ORTEP diagrams and X-ray parameters are provided in the Supplemental Material (Figs. S3, S4 and Tables S4, S5) and Supplementary Data 4/5. The crystal data have been submitted to CCDC with reference numbers 2035165 for **6d** and 2035167 for **6k**. The X-ray studies confirm the structure of compound **6d** as *N-Benzyl-N-phenylethyl 4-nitrophenyl troponyl dihydrapyrazine carboxylic acid*, and of compound **6k** as *N-octyl-N-phenylethyl 4-fluorophenyl troponyl dihydrapyrazinium carboxylate*. The single crystal structure of compound **6k** exhibit that carboxylate *O*-atom is hydrogen bonded with H-atom (O = C–O-----H–N) with octyl amine salt at distance of 1.7 Å (SM, Fig. S3). As shown in Fig. 2, imines of benzylamine (**4a**) and aromatic aldehyde (**3b-3h**) including cinamaldehyde (**3i**) produced respective cATC derivatives **6b/6d-6g/6m-6o**; imines of phenylethyl amine (**4b**) and 4-nitrobenzaldehyde (**3c**) gave cATC (**6c**); imines of 2-butylamine (**4c**) and 4-hydroxybenzaldehyde (**3j**) produced cATC **6 h**; imine of propargyl amine (**4d**) and benzaldehyde (**3a**) produced cATC (**6i**); imine of octylamine (**4e**) with benzaldehyde (**3a**)/4-fluorobenzaldehyde (**3d**) produced respective cATC derivatives **6j** and **6k**. Herein, we successfully synthesized various types of substituted *cyclic-aminotroponimine* derivatives and confirmed their structures by NMR, HRMS, and X-ray studies.

**Mechanistic studies**. We monitored the progress of cATC formation (**6a**) from troponyl amino acid ester (**2**) and imine (**5a**) by NMR, Mass, and UV spectrometer. We recorded time-dependent $^1$H-NMR spectra of ester (**2**) after the addition of TFA (6% in CDCl$_3$) followed by imine (SM, Figs. S5, S6). We noticed that α-

CH$_2$ of ester **2** (δ4.3 ppm) was shifted to δ4.5 ppm (downfield) after the addition of TFA, and then a new peak appeared at δ4.6 ppm (singlet) just after 30 min. The intensity of new peak (δ4.6 ppm) was progressively increasing with respect to time at the expense of downfield shifted peaks (δ4.5 ppm, singlet). Presumably, the significant downfield shift of α-CH$_2$ occurred due to the formation of a new intermediate with TFA, which was further converted into cATC product (**6a**). Thus α-protons of ester **2** are labile under acidic conditions. Time-dependent mass analyses further support the formation of cATC derivative (SM, Figs. S7, S8). Since tropolone-related derivatives are UV–Vis chromophores, we monitored the reaction progress with time by UV–Vis/fluorescence spectrophotometer (SM, Figs. S9, S10). UV spectrum of reactant ester (**2**) exhibits characteristic peaks (~λ$_{250nm}$, ~λ$_{360nm}$ & ~λ$_{420nm}$) and aryl imine (**5a**) at ~λ$_{260nm}$. The UV spectrum of the isolated product (**6a**) exhibits peaks at λ$_{270nm}$, λ$_{370nm}$ & λ$_{460nm}$. We noticed the appearance of a new absorption peak (~λ$_{380nm}$) from reactant **2** after the addition of TFA that further collapses into a new peak (λ$_{460nm}$) with respect to time after the addition of imine (**5a**). The new absorption peaks appeared owing to the formation of a new product as cATC derivative (**6a**). Thus a new intermediate with absorption peak (~λ$_{380nm}$) has been formed before the addition of imine and cATC formation.

We propose the mechanism of cATC product formation as shown in Fig. 3. Previously, we have reported the formation of *cyclic tropolonium cation* (CTL)-intermediate (**2***) followed by the reactive species *ketene* (**2****) from troponyl amino acid ester/ amide derivatives under mild anhydrous acidic conditions[29]. Our time-dependent $^1$H-NMR and UV–Vis spectral results suggest that the reactive intermediate (peak at ~λ$_{380nm}$) could be cyclic tropolium cation (CTL)-intermediate (**2***) which further converts into a reactive intermediate *ketene* (**2****) under acidic conditions owing to labile nature of α-protons of ester **2**. In the literature, *ketene* and *imine* are known to form a cyclo-adduct product (*lactam*)[30,32–34]. It is also reported that *ketene* also forms adduct with *alkene*[35]. Similarly, the *ketene* of ester (**2**) forms a new adduct *lactam* with *imine* (**5**) via [2 + 2] cycloaddition reaction in the same reaction pot. However, we could not isolate the *lactam* (**Int-1**), presumably owing to the reactive *N*-troponyl amide bond. This unstable *lactam* (**Int-1**) is further cleaved and rearranged into another reactive cationic tropolonium lactone (CTL) intermediate (**Int-2**) and an intramolecular alkylaminyl

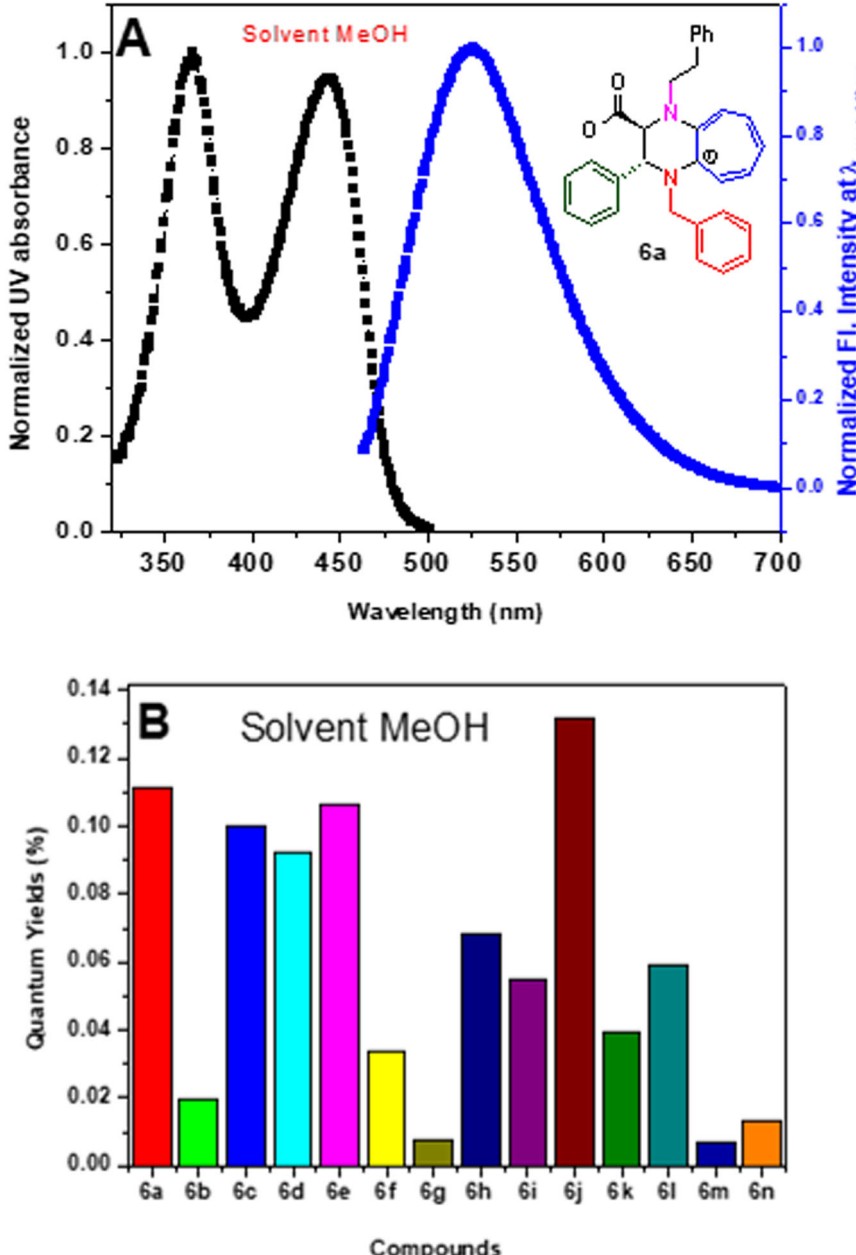

**Fig. 4 Photophysical studies of cATC derivative. A** shows UV–Vis and fluorescence spectra of cATC derivative (**6a**). **B** bar-diagram shows the quantum yield of cATC derivative (**6a-6o**).

ion. In the last, alkylaminyl ion reacts with *troponium* cation via addition-elimination reaction (SNAr) to produce the stable product cATC (**6**). We also performed DFT (B3LYP) calculations and extracted single point energy of reactant, products, and intermediates at ground state using Gaussian software. The details are provided in SM (Figs. S11, S12) and Supplementary Data 6. Their relative energy profile supports the formation of *lactam* intermediate (**Int-1**) followed by lactone intermediate (**Int-2**) which converts into cATC products.

**Photophysical studies**. In the literature, Tropolone and related compounds exhibit characteristic UV–Vis absorption and fluorescence properties owing to π–π*, n–π* transitions and intramolecular charge transfer, though their quantum yield is very low[25,36,37]. We noticed that cATC derivatives are soluble in most of the organic solvents. We assume that cATC derivatives exhibit

strong fluorescence owing to the existence of aminotroponiminium cation species, which induces better delocalization of π–electron and lowers the π–π* electronic transition as compared to the non-cationic troponoid derivatives. We found that the methanolic solution of cATC derivatives was fluorescent (SM, Fig. S13). Herein, we examined the photophysical properties of representative cATC derivatives (**6a-6o**). We recorded the UV–Vis/fluorescence spectra and measured their extinction coefficients and quantum yield in polar solvent MeOH (Fig. 4 and Supplementary Material Figs. S14–S17). In Fig. 4A, the UV–Vis spectrum of cATC derivative (**6a**) exhibits two absorption peaks at the wavelength 370 nm and 460 nm ($\lambda_{abs,370\,nm/460nm}$) while its emission spectrum exhibits only one peak at the wavelength ~525 nm ($\lambda_{em,525nm}$) at excitation wavelength 440 nm ($\lambda_{ex,440nm}$) with Stokes shift ~70 nm. The UV–Vis and fluorescence spectral pattern and Stokes shift of other carboxylates (**6b-6o**) were similar. We also calculated their extinction coefficient and relative

quantum yield with the reference of Coumarin with $\lambda_{ex,440\ nm}$ (SM, Table S6). Their extinction coefficient values varied with substituents in the range of ~14,000–18,000 $M^{-1}\ cm^{-1}$. The relative quantum yield values (~0.1–12%) of cATC derivatives (**6a–o**) are summarized in the bar diagram (Fig. 4B). The dimethoxy-phenyl/naphthyl substituent significantly decreases the quantum yield of respective cATC derivative (**6b/6g**) possibly due to the steric factors. The nitrile-phenyl, fluoro-phenyl and hydroxyl-phenyl substituents also decrease the quantum yield of respective cATC derivatives (**6f/6k/6l/6m**), possibly due to the electron-withdrawing effects and electronegativity. The *fluorobenzene*-cATC derivative (**6m**) has the lowest quantum yield but has a large stokes shift (100 nm) as compared to other cATC derivatives. However, the quantum yield of lone tropolone and aminotropolone compounds is lower (0.1–2.0%) in nonpolar solvents and negligible in polar solvents. The fluorescent enhancement of cATC is mainly owing to the presence of electron-donating groups alkyl-amines and *troponium-cation* charge species at tropone ring, which eases the delocalization of conjugated π-electrons and positive charge. Its carboxylate anion is acting as a counter ion of *troponiun-cation* and enhances charge separations. Lone pairs of *alkyl* substituted diamine extend the conjugation of tropone ring in cATC derivatives that may lower the energy difference ($E_{HOMO}–E_{LUMO}$) compared to unsubstituted tropolone and aminotropone. We theoretically calculated the HOMO-LUMO molecular diagram/energy of cATC derivative (**6a-6o**) in the gas phase by DFT (B3LYP). Their HOMO-LUMO diagrams are provided in the Supplemental Material (Figs. S18–S20). Their MO diagrams show that troponyl dihydropyrazinium and carboxylate residues are involved in HOMO while only aminotroponyl rings are involved in LUMO. Carboxylate and aminotropimine groups are fluorogenic residues in cATC derivatives. In case of nitro derivative (**6d**), nitrophenyl group is involved in LUMO rather than aminotroponyl residue. Thus, nitrophenyl substituent (β-C position of carboxylate) plays a significant role in the fluorescent behavior of cATC **6d** derivative. We also extracted HOMO-LUMO energy gap ($E_{HOMO}–E_{LUMO}$) of **6a-6o**, which was in the range of 2.6–2.8 eV (SM, Table S7). The β-C-phenyl substituents of cATC derivatives influence the quantum yield of synthesized cATC derivatives owing to perturbation of the excited electron relaxation. Thus, troponyl dihydropyrazinium is a fluorogenic residue in cATC derivatives, and their quantum yield depends upon the stereo-electronic nature of phenyl substituents at β-C of carboxylate. Hence, cATC derivatives are strong fluorescent molecules in comparison to tropolone, and the other reported 2-aminotropone/amintroponimine derivatives.

**DNA binding studies**. Polyamine and benzenoid aromatic diamines strongly bind with DNA duplex structure and are considered therapeutic drug candidates[38,39]. Accordingly, we examined interaction of cATC derivatives with DNA duplex structure by UV–Vis/fluorescence spectrophotometer. We recorded the absorption and emission spectra of cATC derivatives (**6a/6e/6j/6n**) with different concentrations of *ct*-DNA duplex at room temperature and pH 7.1. The absorption spectrum of cATC derivative (**6j**) exhibits hyperchromicity ($\lambda_{260nm}$) and isosbestic point ($\lambda_{300nm}$) with the increasing concentration of *ct*-DNA (SM, Fig. S21E). The emission spectrum of fluorescent compound **6j** exhibits a remarkable enhancement in its fluorescence intensity ($\lambda_{ex.440nm}$; $\lambda_{em.520nm,}$) with the increasing concentration of *ct*-DNA (SM, Fig. S21F). Other cATC derivatives **6a/6e/6n** also exhibit similar absorption hyperchromicity (~$\lambda_{260nm}$) and fluorescence enhancement (~$\lambda_{ex.440nm}$, $\lambda_{em.520nm}$) with the increasing the concentration of *ct*-DNA at pH7.0 (SM, Fig. S21). These

results strongly support the interaction of cATC derivatives with *ct*-DNA. Next, we examined the binding mode of representative cATC derivative (**6j**) with *ct*-DNA by competitive assay using DNA intercalating agent, ethidium bromide (EtBr), and the groove binding Hoechst dye. In that assay, the fluorescence intensity of DNA-EtBr (or DNA-Hoechst) complex is quenched by DNA binding agent (intercalative or groove binding)[40–42]. Herein, we recorded the fluorescence spectra of DNA complexes (EtBr:*ct*-DNA/Hoechst:*ct*-DNA) with the increasing concentration of cATC derivative **6j** (SM, Figs. S22, S23). Their spectra show the enhancement in the fluorescence intensity of both the complexes (EtBr:*ct*-DNA/Hoechst:*ct*-DNA) with the increasing concentration of cATC (**6j**). These fluorescence studies reveal that cATC (**6j**) binds with *ct*-DNA duplex without replacing EtBr and Hoechst binding sites. The core structure of cATC, *troponyl-dihydropyrazine*, is a non-planar twisted structure, strictly not favorable for the intercalation with DNA nucleobase pair, but its unique structure, possessing troponium cation, *cyclic*-diamine, and carboxylate functionality probably interact with Hoogsteen sites of nucleobases and the phosphate backbone of DNA at the major groove of DNA duplex structure. Overall, cATC derivatives bind with DNA duplex structure and enhance the fluorescence which could be used as a fluorescent DNA binding agent.

Further, we examined the effect of cATC derivatives (**6a/6e/6j/6n**) on the *ct*-DNA duplex structure by circular dichroism (CD) studies. We recorded the CD spectra of *ct*-DNA with the increasing concentration of cATC derivatives (SM, Fig. S24). The CD spectrum of control *ct*-DNA exhibits the characteristic B-type duplex structure that almost remained unchanged by increasing the concentration of cATC derivatives (**6a/6e/6j/6n**). Next, we examined the binding site of cATC derivative (ligand) at the DNA duplex structure by using AutoDock-Vina, a well-reported computational docking program[3]. We performed docking of cATC derivatives (**6a/6d/6k**) crystal structure (ligand) with dodecamer DNA duplex crystal structure (PDB ID-1BNA) as the receptor. Our docking results strongly support the non-covalent interaction of cATC derivatives (**6a/6d/6k**) at the DNA-minor groove with the binding affinity ~7.0 Kcal/mole, mostly hydrophobic interactions (SM, Figs. S25–7, Table S8). Importantly, the carboxylate group octyl-cATC derivative (**6k**) exhibits hydrogen bonding with guanine residues of DNA. Overall, cATC derivatives non-covalently bind with DNA duplex through the non-intercalative mode without affecting the native duplex structure. Next, we examined the DNA binding affinity of cATC derivatives (**6e/6j/6n**) by gel-electrophoresis. The different concentrations of human genomic DNA were treated with EtBr (control) and cATC derivatives (**6e/6j/6n**) and ran on 1% agarose gel (SM, Fig. S28A–D). The DNA bound cATC derivatives (**6e/6j/6n**) were visible on agarose gel under UV light and the signal was equivalent to EtBr indicating that cATC derivatives bind with DNA and make it visible. Further, the Microscale thermophoresis (MST) technique was used to calculate the binding affinities of cATC derivatives (6e/6j/6n) (20 μM) with varying concentrations of human genomic DNA (40 fM to 75 pM). Hill's coefficient (nH) was recorded as 1.6, 5.8, and 6.3 for **6e/6j/6n**, respectively (SM, Fig. S28E–G), suggesting a higher binding affinity for **6j** and **6n** compared to **6e**.

*cATC derivatives (6n/6j/6e) reduce cell proliferation*. We investigated the effect of nine cATC derivatives (**6b/6d/6e/6f/6i/6j/6k/6l/6n**) on cell viability in cancer cell-line HeLa and transformed normal cell-line HEK293. We treated cells with the varying concentrations of the compounds ranging from 10 nM to 50 μM for 18 h (optimized).

Three compounds, **6e**, **6j**, and **6n** retarded HeLa cell proliferation in a dose dependent manner (Fig. 5A–C) whereas compounds **6b**, **6d**, **6f**, **6i**, **6k**, **6l**, had no or minimal effect on the cell proliferation

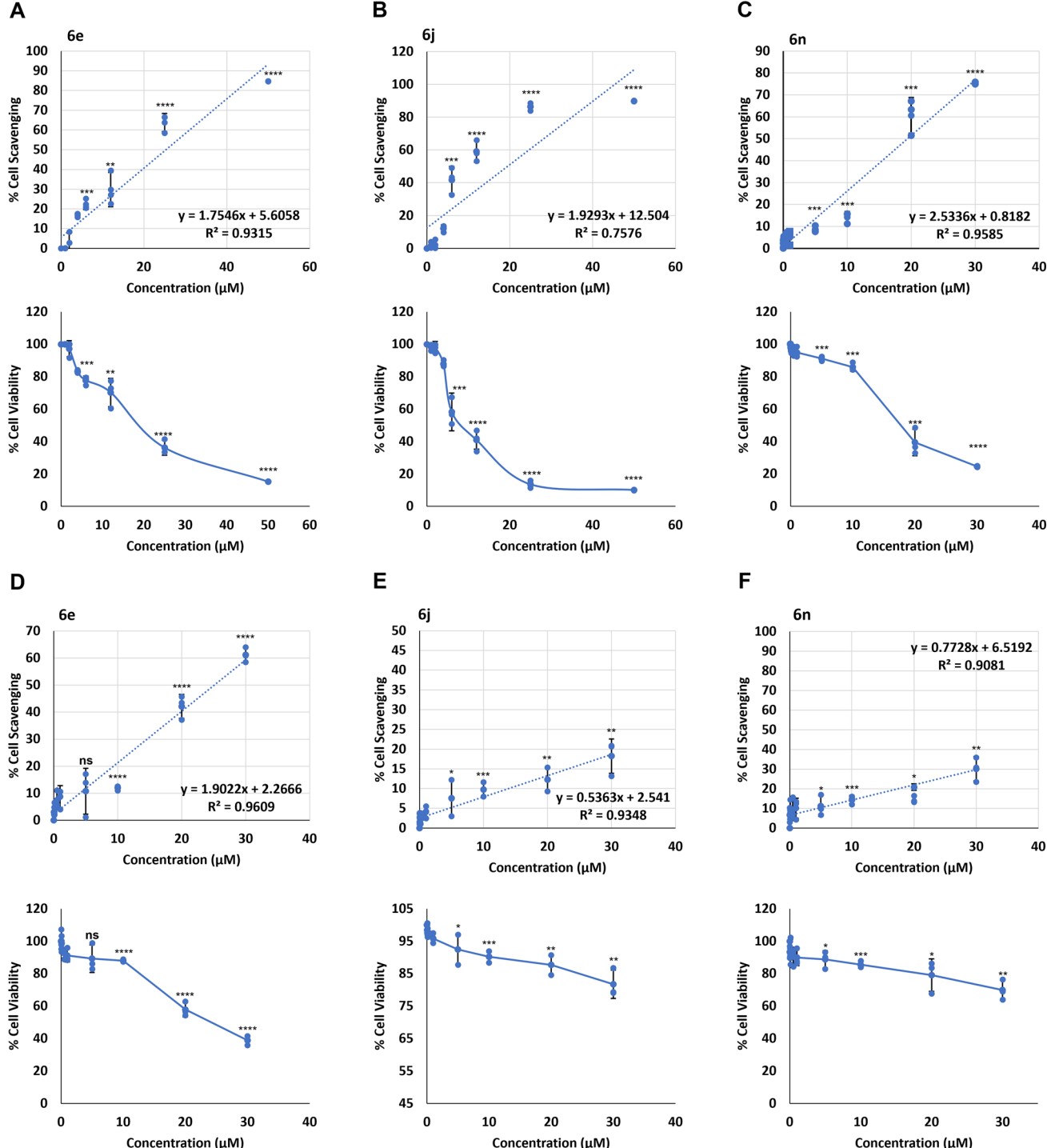

**Fig. 5 Effect of cATC derivatives (6e, 6j, and 6n) on cell growth.** HeLa cells were treated with indicated amount of **6e**, **6j**, and **6n** for 18 h. **A–C** Cell scavenging, and cell viability was determined relative to the cells treated with solvent (DMSO) only; $N = 3$; mean ± SD. **D–F**, Cell scavenging, and cell viability was determined in HEK293 cells relative to the cells treated with solvent (DMSO) only; $N = 3$; mean ± SD; two-tailed unpaired Student's $t$ test was used for calculating $P$ values; ns represents non-significant, * represents $p$-value ≤ 0.05, ** represents $p$-value ≤ 0.01, *** represents $p$-value ≤ 0.001 and, **** represents $p$-value ≤ 0.0001, $N$ represents experiment replicates.

rate (SM, Figs. S29, S30, S32, S33, S35–36). Cell scavenging for **6e** and **6j** at 50 µM was 84.74% and 89.87%, respectively, and for **6n** at 30 µM it was 75.43%. Corresponding cell viability was 15.25%, 10.13% and, 24.56%, respectively. Fluorescence signal became visible inside cells at 12 µM, 12 µM, and 20 µM concentrations for **6e**, **6j**, and **6n**, respectively (SM, Figs. S31, S34, S37). Interestingly, in HEK293 cells we found lesser toxicity, especially for **6j** and **6n**.

Scavenging activity for **6e**, **6j**, and **6n** was 61.1%, 18.23%, and 30.1%, respectively at 30 µM concentration (Fig. 5D–F). Corresponding cell viability was 38.82%, 81.76%, and 69.86%, respectively. The IC$_{50}$ value of all the compounds was at the micromolar (µM) level which is shown in the Supplemental Material (Table S9 and S10). The IC$_{50}$ concentrations for **6j** (19.43 µM) and **6n** (19.40 µM) were lower than **6e** (25.30 µM), which might be a reflection of

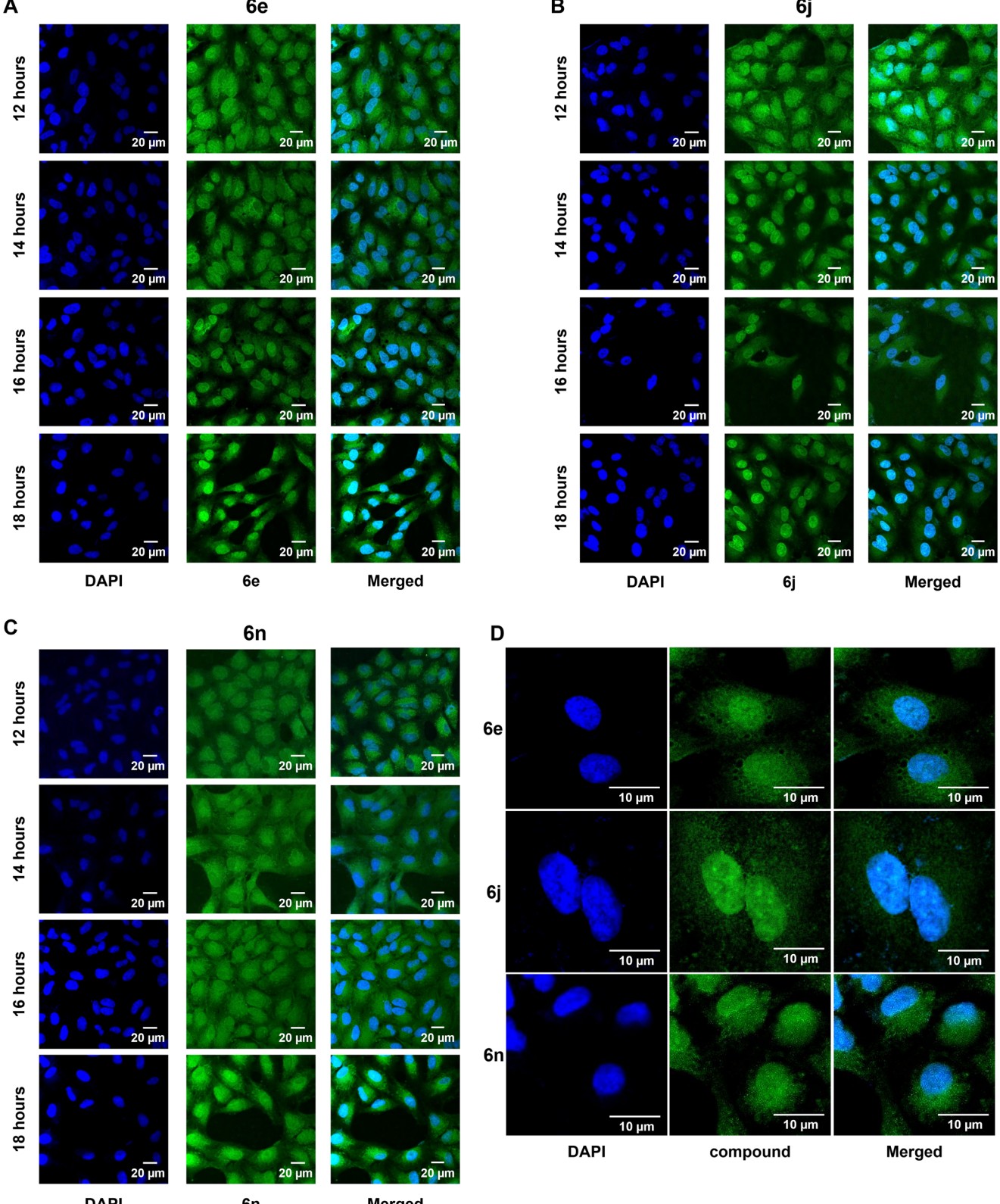

**Fig. 6 Localization of cATC derivatives in cells.** Confocal microscopy images of HeLa cells treated with **6e**, **6j**, and **6n**, and DMSO for 12, 14, 16, and 18 h. **A**–**C**, Middle panel shows green fluorescent signal for **6e** (**A**), **6j** (**B**), and **6n** (**C**) in cytoplasm and nucleus at different time points. Left panel shows DAPI staining for nucleus. Right panel shows merged images. Scale bar is 20 μm. **D** Middle panel shows images of HeLa cells at 60x fold magnification for **6e, 6j**, and **6n** at 18 h. Left panel shows DAPI staining for nucleus. Right panel shows merged images. Scale bar is 10 μm.

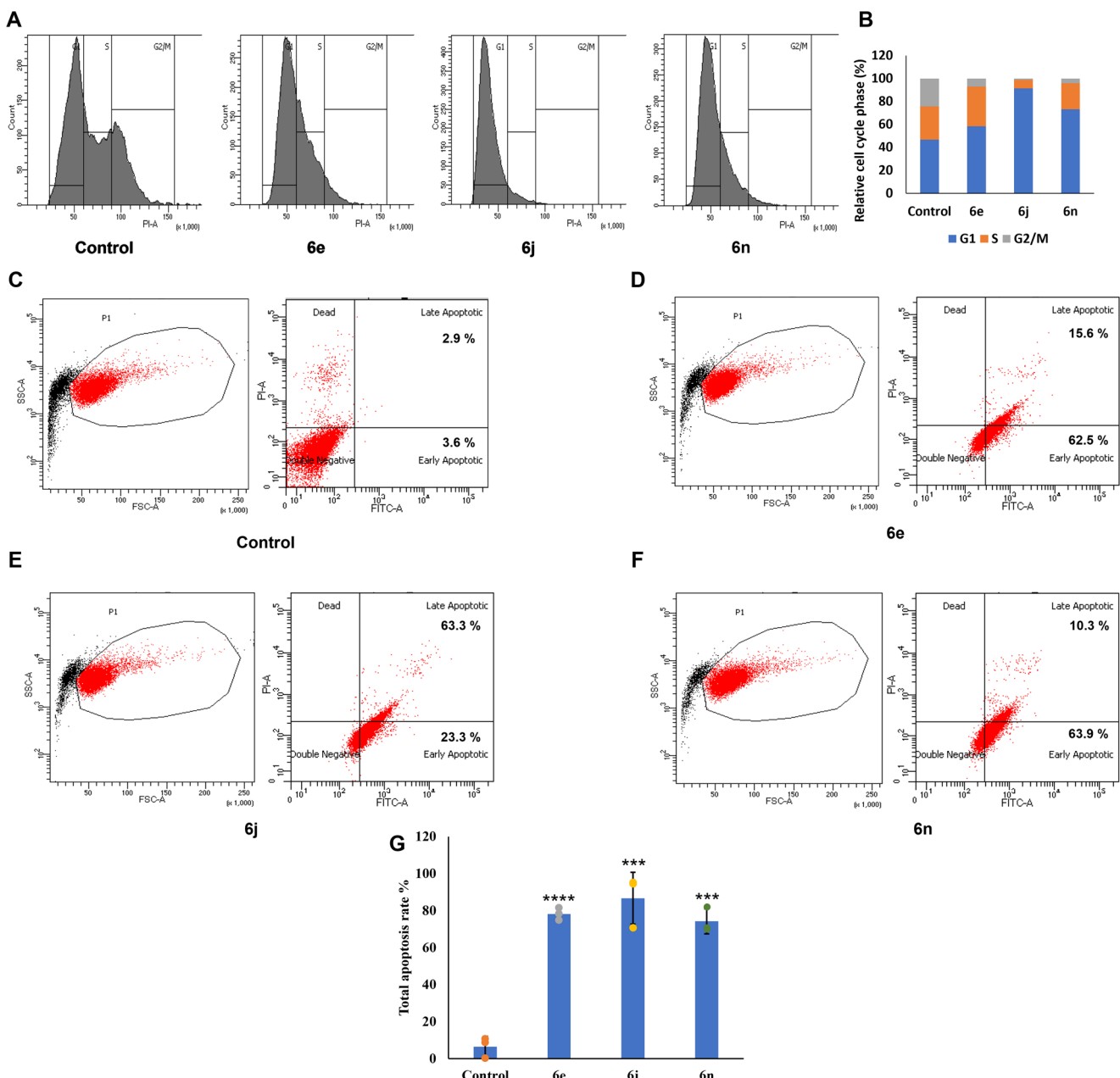

**Fig. 7 Flow cytometric analysis of cell cycle and apoptosis in cells treated with cATC derivatives.** HeLa cells were treated with **6e**, **6j**, and **6n** for 18 h. **A–B** The cell cycle was evaluated by flow cytometry after staining with propidium iodide (PI) for 30 min at 37 °C in dark. **A** Graphs show the cell count in different phases of cell cycle after treatment with **6e, 6j,** and **6n** and control. **B** Histogram shows % cell distribution of HeLa cells in different phases of cell cycle. **C–G** The cells were stained with FITC-conjugated Annexin V (AV) and Propidium Iodide (PI), and analyzed by flow cytometer. The scatter plots for cells unstained (**C**), treated with DMSO (**D**), treated with $IC_{50}$ concentrations of **6e, 6j** and **6n** (**D**, **E**, and **F**, respectively). The early apoptotic cells (FITC-AV$^{+ve}$/PI$^{-ve}$) are shown in lower right quadrant. The late apoptotic cells (FITC-AV$^{+ve}$/PI$^{+ve}$) are shown in upper right quadrant. **G** Bar graph shows total population of apoptotic cells in various samples. $N = 3$, mean ± SD; two-tailed unpaired Student's $t$ test was used for calculating $P$ values; ns represents non-significant, * represents $p$-value ≤ 0.05, ** represents $p$-value ≤ 0.01, *** represents $p$-value ≤ 0.001 and, **** represents $p$-value ≤ 0.0001, $N$ represents experiment replicates.

better binding affinities of **6j** and **6n**. Taken together, the cell viability study demonstrated that out of the nine compounds, **6e, 6j**, and **6n** were the most potent anti-proliferative compounds. **6j** and **6n** showed the minimum effect on normal cells, signifying their therapeutic potential.

*cATC derivatives (6n/6j/6e) mainly localize to the nucleus.* Herein we have shown that cATC fluorescent derivatives bind with *ct*-DNA in the non-intercalating mode. Thus we assumed that cATC

derivatives might bind to the DNA, localizing within the nucleus and affecting the cell division. We studied the localization of fluorescent signals in HeLa cells using $IC_{50}$ concentrations of compounds (**6e, 6j**, and **6n**) at different time points using a confocal microscope. For all the compounds initially (at 12 h), the signal was present in the cytoplasm and the nucleus (Fig. 6A–C). With increasing time, the signal became more intense in the nucleus (Fig. 6A–D). These results suggest that the cATC derivatives (**6e, 6j**, and **6n**) penetrate the cell membrane and localize to the nucleus.

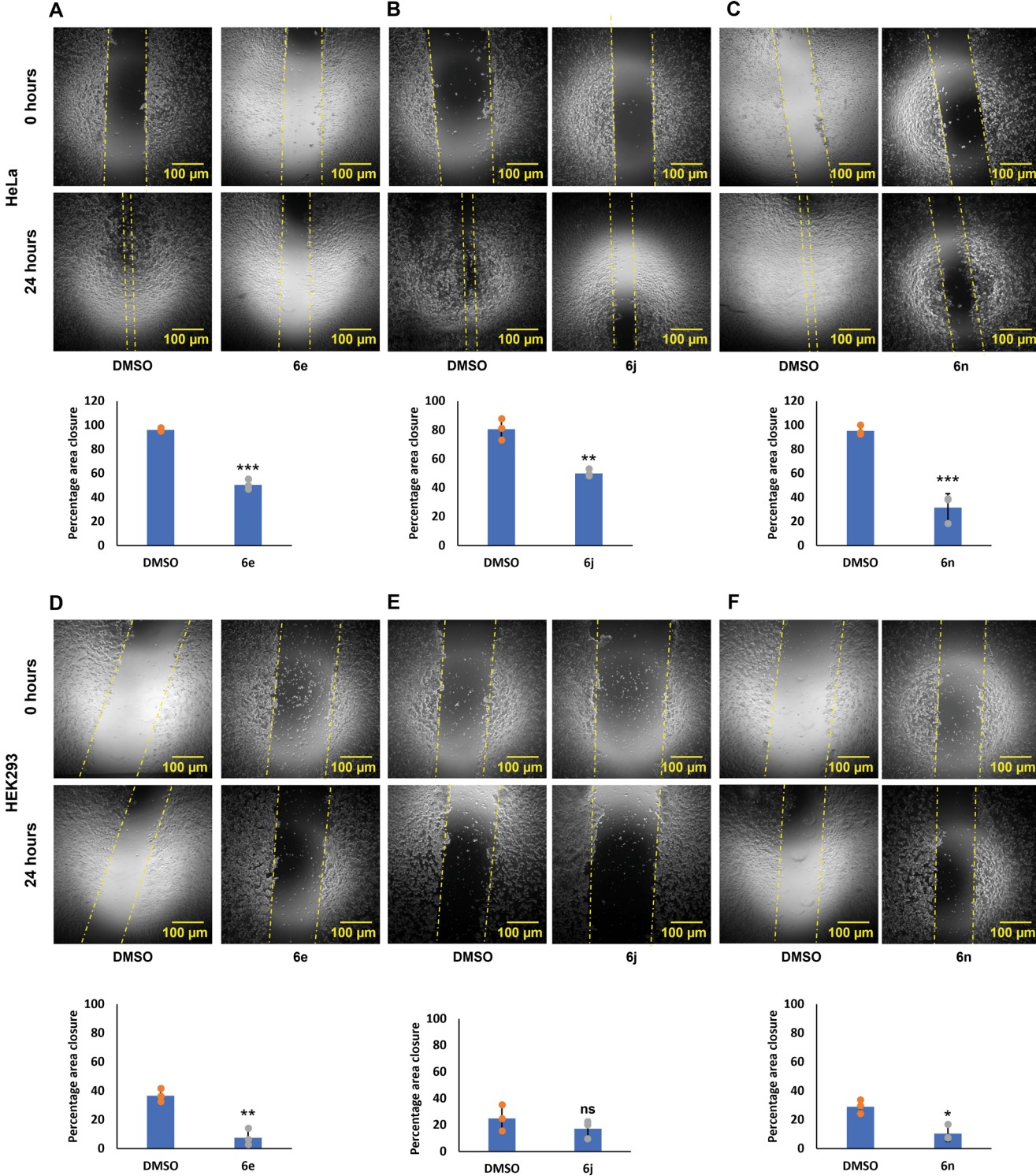

**Fig. 8 Effect of cATC derivatives on cell migration.** Wound healing assay was done in HeLa and HEK293 cells treated with **6e** (18 μM), **6j** (19.23 μM), **6n** (18.92 μM), and DMSO, for 24 h. **A–C** Images of wound at 0 h and 24 h in HeLa cells treated with **6e** (**A**), **6j** (**B**), and **6n** (**C**). Corresponding bar graphs showing percentage of wound area closure in 24 h. **D–F** Images of wound at 0 h and 24 h in HEK293 cells treated with **6e** (**D**), **6j** (**E**), and **6n** (**F**). Corresponding bar graphs showing percentage of wound area closure in 24 h. $N = 3$; mean ± SD; two-tailed unpaired Student's $t$ test; ns represents non-significant, ** represents $p$-value ≤ 0.01, *** represents $p$-value ≤ 0.001 and, **** represents $p$-value ≤ 0.0001, $N$ represents experiment replicates. Scale bar is 100 μm.

*cATC derivatives (6n, 6j, and 6e) arrest cell cycle and induce apoptosis.* We further analyzed the mechanism of effect on the cell proliferation by the cATC derivative **6e, 6j,** and, **6n**. Based upon the localization result, we assumed that on binding with

DNA, the compounds might be triggering the cancer cells to undergo cell cycle arrest followed by apoptosis. Flow cytometric analysis in HeLa revealed that cATC derivative **6e** (18 μM) treatment led to the arrest of 58.33% of cells into G0/G1, 34.61%

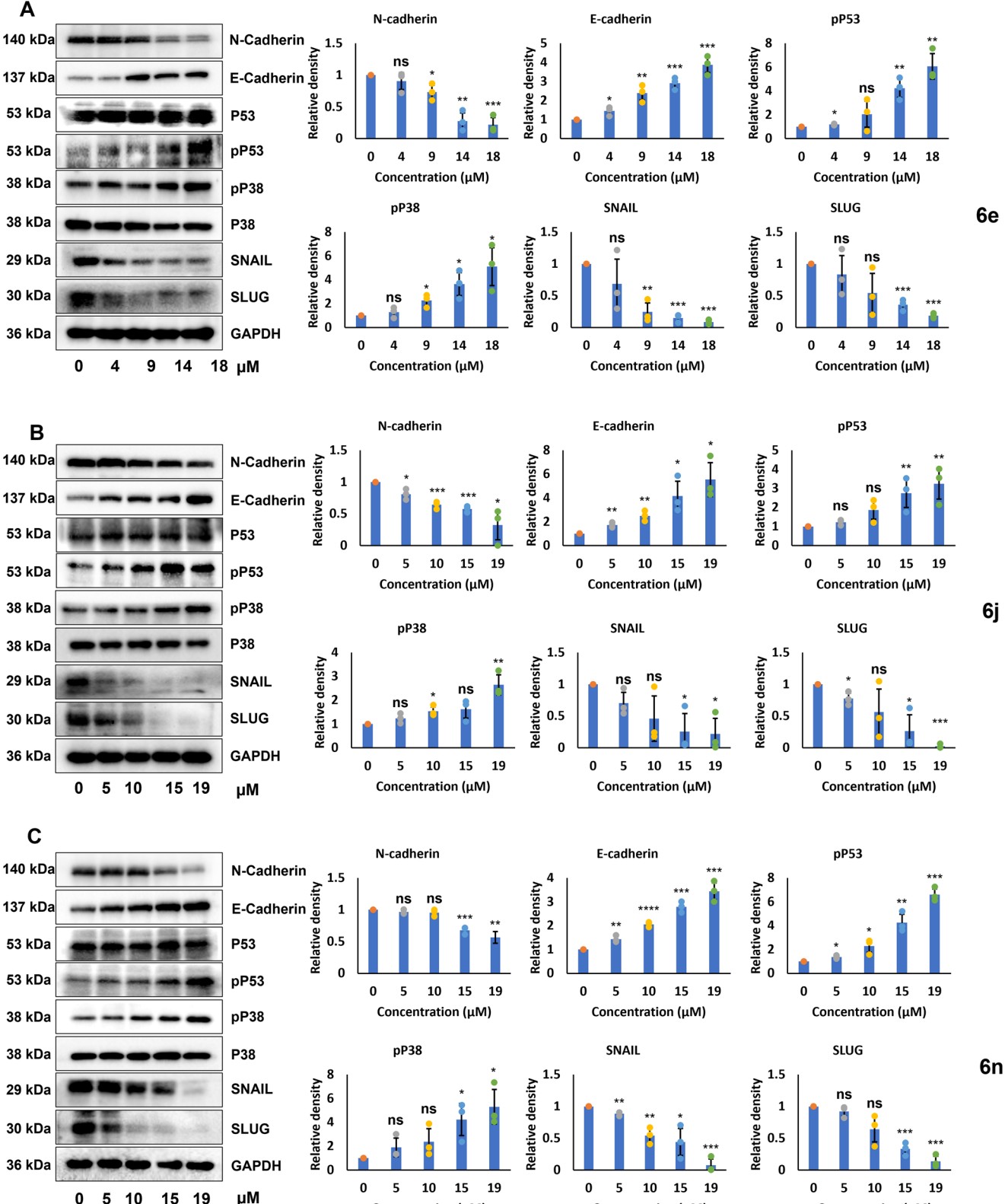

**Fig. 9 cATC derivatives 6e, 6j and 6n activate pro-apoptotic signaling and suppress expression of EMT markers in HeLa cells. A–C** Western blot analysis of cellular lysates prepared from HeLa cells treated with varying concentrations (0–19 μM) each of **6e** (**A**), **6j** (**B**) and **6n** (**C**). Left panel shows Western blot images for E-Cadherin, N-Cadherin, total P53, phospho-P53, phospho-P38, total P38, SNAIL and SLUG. GAPDH was used as loading control. Bar graphs in right panel show mean ± SD average value of three experiments for densitometric analysis relative to GAPDH. Difference in expression level was analyzed compared to 0 μM (DMSO only) using two-tailed unpaired Student's $t$ test. ns represents non-significant, * represents $p$-value ≤ 0.05, ** represents $p$-value ≤ 0.01, *** represents $p$-value ≤ 0.001, **** represents $p$-value ≤ 0.0001, and, $N$ represents experiment replicates.

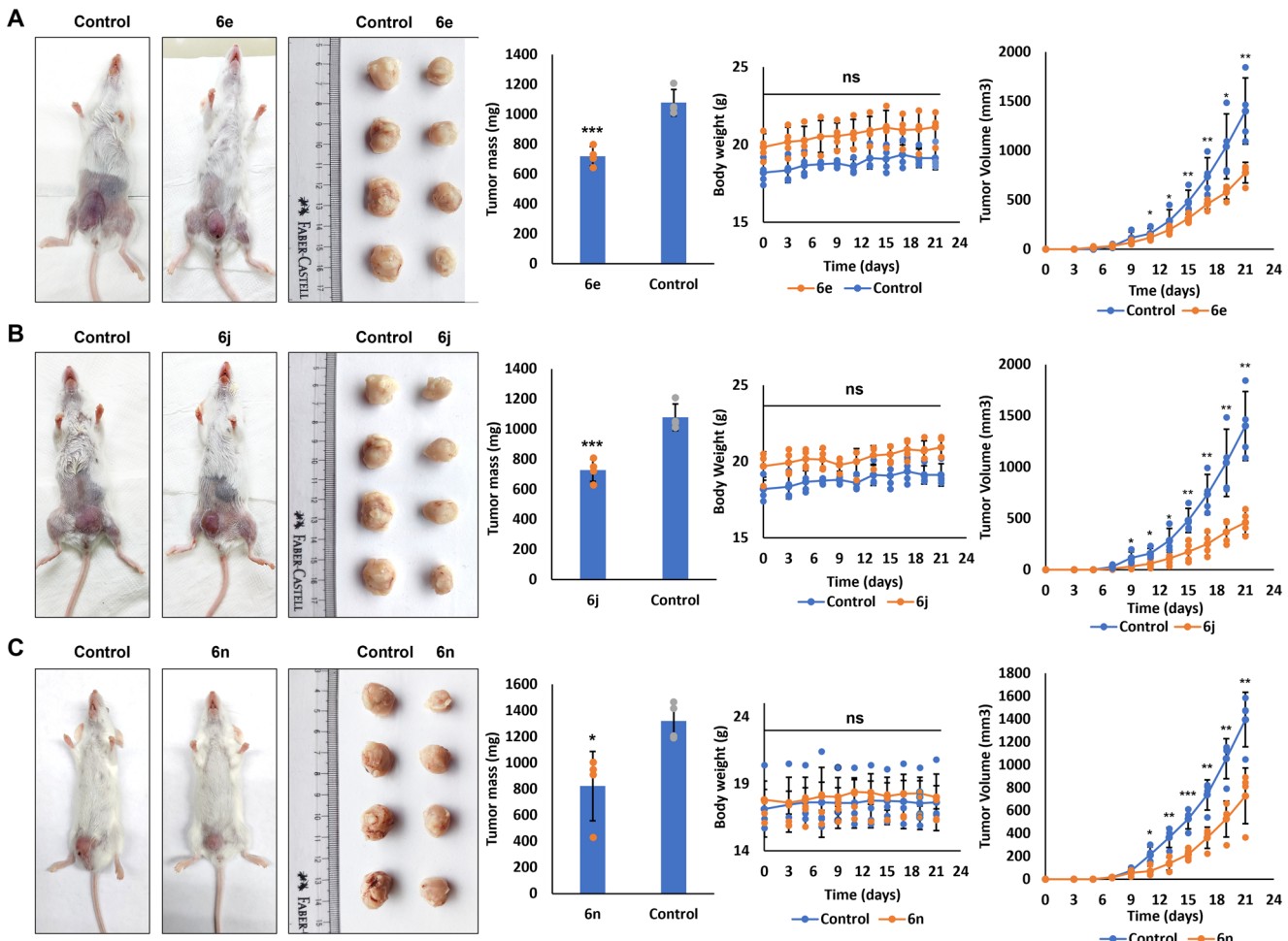

**Fig. 10 cATC derivatives 6e, 6j, and 6n inhibit tumor growth in vivo.** 4T1 cell derived tumor was induced in Balb/C mice and the animals were treated with **6e**, **6j**, and **6n** compounds (1 mg/kg/day, 0.5 mg/kg/day, and 1 mg/kg/day, respectively) starting from day 8 till day 20, post tumor induction. **A–C** The representative images of mice with tumor. Right-hand side images show tumors from mice treated with cATC derivatives **6e** (**A**), **6j** (**B**) and **6n** (**C**) and DMSO (control). Bar graphs (left) and line graph (middle) show average tumor mass and body weight in each group, respectively. Graphs in right show external tumor volume measured every alternate day. Each group had four animals. Values are mean ± SD; two-tailed unpaired Student's *t* test was used for calculating P values; ns represents non-significant, * represents *p*-value ≤ 0.05, ** represents *p*-value ≤ 0.01, *** represents *p*-value ≤ 0.001 and, **** represents *p*-value ≤ 0.0001, *N* represents experiment replicates.

of cells into S, and 7.05% of cells into G2/M phase, respectively. In the case of cATC derivative **6j** (19 μM), the majority of the cells (91.51%) were in G0/G1, 7.23% of cells were in S, and 1.25% of cells were in G2/M phase. Similarly, cATC derivative **6n** (19 μM) treatment led to the arrest of the majority (73.23%) of cells into G0/G1, 22.69% of cells into S, and 4.06% of cells into G2/M phase, respectively. In the control set, we found 46.78% cells at the G0/G1 phase, 28.84% cells at the S phase, and 24.36% cells at the G2/M phase, indicating that cATC derivative treatment of HeLa cells led to suppression of cell cycle progression (Fig. 7A, B). We further evaluated the induction of apoptosis in the HeLa cells, which were treated with cATC derivatives (**6e/6j/6n**) for 18 h. FACS-based detection of fluorescein isothiocyanate (FITC)-labeled annexin V and propidium iodide (PI) signal showed that for **6e**, **6j**, and **6n** compounds, 78.2%, 86.7%, and 74.23% of the cells (respectively), entered into apoptotic phase compared to 6.5% in the control condition (Fig. 7C–G). These findings confirm that cATC derivatives (**6e/6j/6n**) induced apoptosis-mediated cell death.

*cATC derivatives (6n/6j/6e) suppress cell migration.* Cell migration is an integral part of metastasis process and an important

attribute of anti-cancer agents. We examined the effect of **6e** (18 μM), **6j** (19.23 μM), and **6n** (18.92 μM) cATC derivatives on the migration of HEK293 and HeLa cells in a scratch wound. Images of scratch areas from 0 h to 24 h showed that all the three compounds inhibited cell migration significantly in HeLa cells (Fig. 8A–C), and the maximum inhibition was observed in **6n**. We observed that the extent of effect varied in two cell types. cATC derivatives (**6e/6j/6n**) filled the scratch wound to the significantly lesser extent in the HEK293 cells than in HeLa cells (Fig. 8D–F). Our results suggest that the effect of all the compounds is more prominent in the cancer cell line than in the normal transformed cells.

*cATC derivatives (6e/6j/6n) activate p53-p38 pathway and affect EMT markers.* In order to figure out the mechanism of apoptosis induction, we analyzed the effect of different concentrations of **6e** (0 μM, 4 μM, 9 μM, 14 μM and 18 μM), **6j** (0 μM, 5 μM, 10 μM, 15 μM and 19 μM), and **6n** (0 μM, 5 μM, 10 μM, 15 μM and 19 μM) compounds on p38-p53 pathway. HeLa cells showed a gradual increase in phospho-P53 and phospho-P38 levels with the increasing concentration of cATC derivatives (Fig. 9A–C).

In consistence with the scratch assay results, treatment of HeLa cells with **6e**, **6j**, and **6n** led to an increased expression of E-Cadherin, and it decreased expression of EMT markers (N-Cadherin, Snail, Slug) in a dose-dependent manner (Fig. 9A–C). Together these results suggest that **6e**, **6j**, and **6n** have anti-cancerous properties, which they execute by suppressing cancer cell metastasis, possibly by affecting p53 activation and EMT signaling pathway.

**cATC derivatives (6n/6j/6e) tropolone derivatives in vivo tumor xenograft model**. To understand the potential and tolerance of **6e**, **6j**, and **6n** compounds using in vivo system, we used Balb/C 4T1 cell line xenograft mouse model. Treatment was initiated after tumors reached a mean value of 20.47 mm³ (day 8) and treated with **6e**, **6j**, and **6n** at a dosage of 1 mg/kg/day, 0.5 mg/kg/day, and 1 mg/kg/day, respectively. Body weight and activity were observed three days post injection, every alternative day. At the end of the study, mice in the treatment groups showed no significant difference in body weight than the control group (Fig. 10A–C). They also exhibited normal behavior/activity when compared to the control group. Based on this study, we deduced that these compounds are well tolerated by the animals.

We then evaluated the anticancer efficacy. At the end of the study, mice were euthanized, and the tumors were resected and weighed. We observed a significant tumor size reduction with compounds (Fig. 10A–C), which validated our findings from in vitro systems.

## Discussion

Troponoids have wide-range of bioactivities such as *antibacterial, antifungal, antiviral, insecticidal,* and *anticancer*[5,6]. Natural troponoid, *Colchicine*, is a tubulin-destabilizing agent and exhibits antimitotic activity[43,44]. Humulene and Eupenifeldin exhibit antitumor activities[45]. β-Thujaplicin reduced proliferation in MCF7 cells, suppressed cancer stemness in glioma U87MG, induced cell cycle arrest, apoptosis, and DNA methylation via DNMT1 and UHRF1 in colon cancer, and restricted liver cancer growth by apoptosis and cell cycle arrest[46–50]. It decreased the myofibroblast as well as various EMT markers in oral submucous fibrosis (OSF) by inhibiting the binding of SNAIL to the E-box region in α-SMA promoter[51]. Our scratch assay and Western blot data show that **6e**, **6j**, and **6n** inhibit the migration of HeLa cells and decrease the expression of EMT markers like N-Cadherin, SNAIL, and SLUG (Uncropped data are provided in Supplementary Data 7). The effect of cATC derivatives (**6e**, **6j**, and **6n**) on the inhibition of scratch filling can be the cumulative effect of the inhibition of proliferation and migration. Our result of cell viability depicts that **6j** and **6n** are less toxic to HEK293 compared to HeLa cells, which is similar to observations made by treatment with natural compounds β-Thujaplicin in breast cancer cells, colon cancer and OSF[52]. Upon treatment of HeLa cells with our compounds (**6e**, **6j**, and **6n**) we detected an increase in proapoptotic markers like P53 and P38 which is in accordance with the other studies with natural tropolone derivatives[53–55]. Similarly, synthetic derivative alpha naphthyl tropolone induced P53/p-mTOR/p-AKT signaling which increased caspase 3/7 activity in leukemia cells[11].

Bioactivities of tropolonoids mainly depend-upon the metal-complexation with metalloenzymes due to tropolone's carbonyl and hydroxyl groups[1,2]. The synthetic analogs of tropolones such as *aryl-tropolones, aminotropones* and *aminotroponimines* also exhibit metal complexing properties. β-phenyl-tropolone synthetic analogs of *Thujaplicine*, are potent inhibitors of metalloenzymes and inhibit histone deacetylases activities[11,56,57]. Other synthetic troponoids and aminotroponimines derivatives exhibit better metal-chelating properties as compared to tropolone and 2-aminotropone derivatives owing to the *diamines'*

functionality[14]. Our derivatives are non-benzenoid pyrazine derivatives; they do not have tropolone's functionality therefore, we do not expect chelating properties with metalloenzyme. However, various benzenoid pyrazine derivatives bind with DNA duplex through intercalative mode[44,58]. As expected, our photo-physical and gel electrophoresis studies revealed that **6e**, **6j**, and **6n** bind to DNA duplex. Moreover, HeLa and HEK293 cells showed clear localization in the nucleus in a time-dependent manner. So far, only indirect indications were reported showing the association of tropolone derivatives with DNA.

Fluorescence-based traceability and penetrance in cells are two crucial features of drug candidates. Generally, natural troponoids are permeable to cell-membrane[52]. Wakabayashi has reported that an increase in the number of methylene groups in 2-aminotropone derivatives enhanced not only the cell membrane permeability but also the cytotoxicity[59]. However, cATC derivatives even comprising long-chain hydrocarbons are efficiently permeable to cells and exhibit minimum cytotoxicity to the normal cells. Our study revealed that cATC derivatives very efficiently penetrate cell membranes, which could be owing to the formation of the bipolar structure containing alkyl and aryl substituents at amino troponyl carboxylate. Tropolone and aminotropone show weak fluorescence in the nonpolar environment and do not exhibit fluorescence properties in polar environment which makes traceability challenging in the cells. Our cATC derivatives show stronger fluorescence even in the polar environment owing to the delocalization of conjugated troponium-cation charge comprising electron-donating alkyl-amines. Fluorescent signal was clearly evident in confocal microscopy-based images of cells, making it traceable during experimentation.

As per our knowledge, this is the first study to show in vivo effect of the cATC derivatives in a mouse model, suggesting that these can be explored as anti-cancer drug candidates. No weight loss or diminished activities in mice suggests minimum side effects on the normal cells. Long-term survival studies can further ascertain the therapeutic value of these cATC derivatives. Moreover, cancer-specific studies can be done to ascertain the extent of the anti-cancer effect of cATC derivatives.

## Conclusions

Overall, we have successfully synthesized new $C^{aryl}/N^{alkyl}$-substituted *cyclic*-aminotropiminium carboxylate (cATC) derivatives by developing the unique synthetic method of C-C bond formation. The structure of cATC derivatives is well supported by single crystal X-ray studies of representative compounds. These cATC derivatives are fluorescent even in polar organic solvents, with a quantum yield of ~10–12%, much higher than tropolone and aminotropones. Importantly, cATC derivatives bind with DNA duplex structure in non-intercalative modes. We found nuclear localization and antitumorigenic effect of three cATC derivatives, *m-Tolyl*-cATC, *N-Octanyl*-cATC, and Styrenyl-cATC on the cancer cell line. These cATC derivatives retard the proliferation and migration of cancer cell line (HeLa) more than the transformed normal cells (HEK293). These compounds induce cell death via the activation of p38-p53 mediated apoptosis. They show anti-metastatic potential through the suppression of EMT markers. In the xenograft-based mouse model, these cATC derivatives reduced tumor growth. This study reports the synthesis of DNA-binding fluorescent $C^{aryl}$-$N^{alkyl}$-*cyclic*-amino-troponiminium derivatives which have antitumor potential and can be explored as therapeutic agents.

## Materials and methods

**Cell culture**. HeLa and HEK293 cells were purchased from cell repository, NCCS, Pune, India and were cultured in DMEM (Gibco, USA) supplemented with 10%

FBS (Gibco) and 1X PSA (Himedia, India). 4T1 cells (ATCC, USA) were cultured in RPMI (Gibco) supplemented with 10% FBS (Gibco) and 1X PSA (Himedia). The cells were maintained in a humidifier at 37 °C temperature and 5% $CO_2$.

**Cell viability assay**. MTS assay (CellTiter 96® AQueous One Solution Cell Proliferation Assay (Promega, USA) was used to determine the cell viability. HeLa and HEK293 cells (20,000 cells in each well) were plated in 96 well plate and treated with **6e**, **6j**, and **6n** varying concentrations of the compounds ranging from 10 nM to 50 μM for 18 h. The assay was carried out according to the manufacturer's instructions. Finally, the plate was read in Varioskan™ LUX multimode microplate reader (Thermo Fisher Scientific) at 490 nm.

Cell scavenging activity was calculated using the following formula:
$OD_{(490nm)}$ (control) − $OD_{(490nm)}$ (Test)/$OD_{(490nm)}$ (control) × 100%
Cell Viability of the cell was calculated using the following formula:
$OD_{(490nm)}$ (Test)/$OD_{(490nm)}$ (control) × 100%

**Microscale thermophoresis**. DNA was isolated from $10 \times 10^6$ HeLa cells by QIAmp DNA mini Kit (Qiagen, Germany). cATC derivatives (6e, 6j, and 6n dissolved in a mixture of 1X PBS containing 1% DMSO) and DNA ranging from a concentration of 40 fM to 75pM (resuspended in 1X TE with 0.05% Tween-20), was used for analyzing the interaction of cATC derivatives with DNA. The mixture was incubated at room temperature for 2 h. 15 μl of the above mixture was loaded in capillaries and binding affinity was measured using the NanoTemper instrument followed by analysis through NanoTemper analysis 1.2.231 software.

**Scratch assay**. HeLa and HEK293 cells ($0.25 \times 10^6$) were seeded into 12-well plates to grow in a monolayer for 24 h. A Scratch was made in the middle of the well using a sterile 200 μl pipette tip. The detached cells were removed by washing with 1x DPBS. The cell monolayers were treated for 18 h with 19 μM of **6n** and **6j**, and 18 μM of **6e** with 10% DMEM. DMSO with 10% DMEM was used as a control. Images were collected at 0 h and 24 h under an inverted microscope (Ziess, Germany) at (4×) magnification. Cell migration was analyzed using ImageJ (NIH) software.

**Cell cycle analysis**. HeLa cells were cultured in 6 well plates with initial density 0.5 $\times 10^6$ cells/well. These cells were treated with the $IC_{50}$ concentration of **6n**, **6j**, and **6e** compounds for 18 h. The cells were washed with 1X PBS and fixed with 70% ethanol. After washing, cells were resuspended in 500 μl PBS containing Propidium Iodide (PI) (500 μg/ml), RNase A (1X), and Triton X-100 (0.1%), and incubated for 30 min in the dark. Finally, the cells were analyzed by flow cytometry (BD Biosciences, CA, USA). The relative proportions of cells with DNA content G0–G1 (2n), S phase (>2n but <4n), and G2/M phase (4n) were acquired and analyzed using CellQuest Pro software (BD Biosciences).

**Apoptosis analysis**. The apoptotic effect of **6e**, **6j**, and **6n** compounds on cancer cells was detected using FITC Annexin V Apoptosis Detection Kit I (BD Pharmingen™, NJ, USA) according to the manufacturer's protocol. $0.5 \times 10^6$ million HeLa cells were seeded in a 6-well plate and treated with $IC_{50}$ concentrations for 18 h. Harvested cells were incubated with Annexin V FITC and PI PE-conjugated antibody and apoptosis was evaluated using FACSCalibur (BD Biosciences, CA, USA) flow cytometry. Data were analyzed using CellQuest Pro software (BD Biosciences).

**Cytochemistry**. HeLa cells (60,000) were grown onto a coverslip (Coverglass for cell Growth™, Fisherbrand, USA) in a 24-well plate. The cells were treated with **6e**, **6j**, and **6n** compounds at their $IC_{50}$ dose for 12 h, 14 h, 16 h, and 18 h. Thereafter, cells were fixed in 4% paraformaldehyde (Himedia), washed with ice-cold 1X PBS, permeabilized with 0.25% Triton X-100 (Sigma), and washed with 0.1% PBST. The cell was counter stained with 49,6-diamidino-2-phenylindole, dihydrochloride (DAPI, Invitrogen) and mounted using one drop of ProLong® Gold antifade reagent (Invitrogen) on a slide. Images were acquired by fluorescence microscope (Leica SP8 TCS) and analyzed using ImageJ (NIH).

**Western Blotting**. Cell lysates were prepared from HeLa treated with compounds **6e**, **6j**, and **6n** in ice-cold RIPA buffer (Thermo Scientific, USA), and supplemented with protease and phosphatase inhibitor cocktail (Thermo Scientific, USA). Protein quantification of cell lysates was done using BCA reagent (Thermo Scientific). 20 μg of protein samples were separated on a 12% SDS-PAGE and subsequently transferred onto a PVDF membrane (Merck Millipore, USA). The blots were probed with specific antibodies for p38 (1:1000, #9212, Cell Signaling Technology, USA), phospho-p38 (1:1000, #9211; Cell Signaling technology), p53 (1:1000, #9282, Cell signalling technology), phospho-p53 (1:1000, #9284, Cell Signaling technology), snail (1:1000, #C15D3, Cell Signaling technology), slug (1:1000, #C19G7, Cell signalling technology) and GAPDH (1:5000, #ABM22C5, Abgenex, India) followed by incubation with HRP tagged anti-mouse IgG secondary antibody (Abgenex) for GAPDH and HRP tagged anti-rabbit IgG secondary antibody (Abgenex) for p38, phospho-p38, p53, phospho-p53, snail and slug. The chemiluminescence signal was detected in ChemiDoc XRS + (Bio-Rad) using SuperSignal™ West Femto reagent (Thermo Scientific).

**Xenograft tumor growth**. All animal experiments were approved by the Institutional Animal Ethics Committee, NISER, India (AH-214). Female BALB/c mice (aged 6–8 weeks, 18–20 g) were injected with $1 \times 10^6$ 4T1 breast cancer cells into the mammary fat pad. After one week, **6j** (0.5 mg/kg), **6e** (1 mg/kg) and **6n** (1 mg/kg) compounds were injected intraperitoneally in mice at the interval of every 24 h for a period of 21 days. Thereafter, the mice were sacrificed and the xenograft tumors were removed, weighed, and photographed.

## Data availability
All data generated during this study are included in this article and Supplementary Information Experimental procedure, X-ray data, HOMO-LUMO related data, UV–Vis and fluorescence spectra, DNA binding are provided in the Supplementary Material. NMR, HRMS and FT-IR of newly synthesized compounds are provided in the supplemental data 1. Single crystal X-ray data as cif files of cATC compounds (**2/6a/6d/6k**) are provided in the Supplemental data files 2–5 Computational data of reaction mechanism intermediates are provided in Supplemental data 6. Western blot uncropped images of all 3 rounds are provided in Supplemental data 7.

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

## Acknowledgements

This work was supported by the Department of Biotechnology, Government of India, (Grant No BT/PR26143/GET/119112/2017). University grant commission (UGC) and National Institute of Science Education and Research (NISER) provided JRF/SRF fellowships to BBP and SAP, respectively. We thank Animal House and Flow cytometry facility, School of Biological Sciences, NISER-Bhuaneswar. We thank Ananya Palo and Ramchandra Soren for helping with Western blot, and UV-Fluorescence spectra data collection. We thank Saswat K, Pati (summer intern), and Deepak K. Panda for helping with theoratical studies.

## Author contributions

B.B.P. has performed synthesis of cATC derivatives, generated their characterization data (NMR, HRMS, IR, UV–Vis, Fluorescence spectra and X-ray data), executed computational studies and compiled manuscript. S.A.P. has performed Electrophoresis, Microscale thermophoresis, Cell Viability assays, Cell migration assay, Apoptosis assay, Immunocytochemistry, Confocal Microscopy, Cell cycle analysis, Western blotting, in vivo experiments, data curation, analysis, and writing original draft. N.K.S and M.D. have conceptualized the whole project, designed the above experiments, planned, analyzed data, supervised experiments, reviewed/edited the manuscript, used funds and lab resources.

## Competing interests

The authors declare no competing interests.
