## [Peer Review File · Communications Chemistry]

Reviewers' comments:

Reviewer #1 (Remarks to the Author):

This manuscript describes a potentially interesting multidisciplinary work that comprises chemical synthesis, mechanistic and photophysical studies and biological validation of the synthesized zwitterionic compounds. However, several major changes are required to fulfill the publication criteria:

1. The manuscript requires extensive editing of English language.
2. The reported mechanistic analyses rely on qualitative arguments and the proposed mechanism (Figure 3) is plausible but speculative. A computational study would improve significantly the understanding of the nature on reaction intermediates Int-1 and Int-2. Do IR studies permit to detect ketene intermediate 2?
3. The photophysical studies of compound 6a reported in Figure 4 appear to be incomplete. In addition, which method to calculate the HOMO-LUMO gap has been used?
4. All compounds 6 are described in the Supporting Information as viscous oils, liquids or solids with colors ranging from light yellow to black (!). However, adducts 6a, 6d and 6k have been crystallized and the respective CCDC files are reported. Have the authors measured the melting points of these compounds?. Are adducts 6 really viscous oils or liquids?
5. Many NMR spectra include impurities. Additional purification is required.

Reviewer #2 (Remarks to the Author):

The author plans to use Caryl-Nalkyl-substituted cyclic-aminotroponiminium carboxylate (cATC) derivatives to verify the anticancer effect in Hela cells. The data is very abundant, but I think the author should present the data more logically, and the key conclusions should be pointed out in each paragraph of the resulting chapter, otherwise, the whole article will be complicated and lengthy. In addition, I still have some questions as follows:

1. The title of Figure 1 is missing, please confirm. Also, Scheme 1 should be part of the figure, it is recommended to rename and insert a figure legend.
2. In Figures 9A-9C, the data of unstained, single-stained PI or single-stained AV do not need to be placed in the results, only the control group, 6E, 6J and 6N are shown. In addition, how many times have the apoptosis results of flow cytometry been repeated? It is recommended that each group be repeated at least three times and statistical analysis is performed before the presentation.
3. In Figure 10, the position of the DMSO group in the histogram is suggested to be placed to the left as in the DMSO group in the result of the wound healing assay.
4. In Figure 11, the doses of cATC derivatives 6E, 6J, and 6N displayed on the resulting graph are inconsistent with the text, please correct this. Also, how are these doses screened out?
5. In addition to SNAIL and SLUG, are there other indicators representative of the EMT phenomenon that can simultaneously verify the inhibitory effect of cATC derivatives 6E, 6J, and 6N?
6. In the discussion section, hinokitol and β -Thujaplicin should be the same intervention, why the effects be explained separately?
7. According to the present data, please compare the efficacies of the three cATC derivatives (6e, 6j, and 6n) in the DNA binding affinities and anti-tumor activity.
8. Authors showed that cATC derivatives (6e, 6j, and 6n) induced apoptotic cell death by FACS based detection of FITC-labelled annexin V and propidium iodide staining. The population was not significantly increased in positive control PI group in figure 9B. On the contrary, an increased cell population stained with annexin V was observed in the positive control PI group. Authors should further confirm the data and repeat this experiment again.

9. In this study, authors conducted migration assays using the scratch assay and revealed that cATC derivatives reduced the migratory ability of HeLa cells after 24 h treatment. Simultaneously, cATC derivatives inhibited cell proliferation and induced apoptosis after 18 h treatment. What is the concentration of cATC derivatives displayed to inhibit cell migration in figure 10? It is unable to rule out the possibility that decreased cell numbers in the area of the scratched wound are primarily due to cell death rather than inhibiting the migratory abilities. Accordingly, the inhibitory effects on cell migration may be overestimated. To perform the migration assay, authors have to consider the doubling time of cells and the effect on growth inhibition.

10. Authors used different cancer cell types in vitro and in vivo. Please explain the reason.

11. Figure 2B is of poor quality. Please provide a replacement with high resolution. In the text, the authors describe that UV-Vis spectra and fluorescence spectra of 6a are shown in figure 5, while 6j was displayed in the legend. Similar errors are found. There is no description of the results shown in figure 7D to 7H. In the figure legend of figure 7, it is stated that "d-f, The cell cycle was evaluated by flow cytometry after staining with propidium iodide (PI)...." However, it seems that the results are cell viability and scavenging. The results of cell cycle are described in other paragraphs which are shown to display in Figure 9 a-h. Again, the results displayed in Figure 9 are actually the examination of apoptosis. Please check.

12. Please show the error bars (standard deviation) and statistical difference if there are three independent tests.

Reviewer #3 (Remarks to the Author):

The manuscript having good novelty and highly significant work for the community with great application as anticancer activity. I recommend this manuscript after the correction of some flaws in the manuscript like

1. In scheme 1, give the exact amount of imine should be added as place of "excess" for troponyl dihydropyrazine.
2. In scheme 1, what is the difference between the structure of 6c and 6d, as it can be seen both the structure has been same.
3. It should be more appropriate if the author should have discuss the mechanism with FRET or PET way.
4. Also given the FTIR spectra in the supporting information.
5. Add some more references like doi.org/10.1021/ja3103007, doi.org/10.1002/jhet.4357 etc.

Rebuttal Letter (Answers to Reviewers' Comments)

Reviewers' comments:

Reviewer #1 (Remarks to the Author):

This manuscript describes a potentially interesting multidisciplinary work that comprises chemical synthesis, mechanistic and photophysical studies and biological validation of the synthesized zwitterionic compounds. However, several major changes are required to fulfill the publication criteria:

Comment 1. The manuscript requires extensive editing of English language.

Answer: Thank you for pointing out. We have edited language.

Comment 2. The reported mechanistic analyses rely on qualitative arguments and the proposed mechanism (Figure 3) is plausible but speculative. A computational study would improve significantly the understanding of the nature on reaction intermediates Int-1 and Int2. Do IR studies permit to detection ketene intermediate 2?

Answer: Thank you for your valuable suggestions. We have modified the reaction mechanism according to the DFT calculation in the gas phase. Details are provided. We could not detect lactam formation (Int-2) by IR.

Comment 3. The photophysical studies of compound 6a reported in Figure 4 appear to be incomplete. In addition, which method to calculate the HOMO-LUMO gap has been used? Answer. Thank you for pointing out. We have recorded UV-Vis/fluorescence spectra, absorptivity, and quantum yield of 6a-6o. Now we have calculated their HOMO-LUMO using DFT, B3LYP, basic set 6-31G. The details are provided in the Supplemental Material. Comment 4. All compounds 6 are described in the Supplemental Material as viscous oils, liquids or solids with colors ranging from light yellow to black (!). However, adducts 6a, 6d, and 6k have been crystallized and the respective CCDC files are reported. Have the authors measured the melting points of these compounds?. Are adducts 6 viscous oils or liquids? 2 Answer: Most of cATC derivatives are gummy in nature. We crystallized a few compounds at low temperatures. Now we have provided the melting point of solid compounds. Comment 5. Many NMR spectra include impurities. Additional purification is required.

Answer: We have further purified and removed impurities by column. Old NMR of impure compounds are replaced with purified one. (See yellow highlighted in the Supplemental Material.

Reviewer #2 (Remarks to the Author):

The author plans to use Caryl-Nalkyl-substituted cyclic-aminotroponiminium carboxylate (cATC) derivatives to verify the anticancer effect in Hela cells. The data is very abundant, but I think the author should present the data more logically, and the key conclusions should be pointed out in each paragraph of the resulting chapter, otherwise, the whole article will be complicated and lengthy.

Answer: We have concluded after the results after each sections. We have underlined the conclusions. In addition, I still have some questions as follows:

Comment 1: The title of Figure 1 is missing, please confirm. Also, Scheme 1 should be part of the figure, it is recommended to rename and insert a figure legend.

Response: We have provided the title in Fig 1 (line 90). As suggested the scheme 1 has been shown as Fig. 2 (line 142). These line numbers is extracted from Manuscript marked up

Comment 2: In Figures 9A-9C, the data of unstained, single-stained PI or single-stained AV do not need to be placed in the results, only the control group, 6E, 6J and 6N are shown. In addition, how many times have the apoptosis results of flow cytometry been repeated? It is recommended that each group be repeated at least three times and statistical analysis is performed before the presentation

Answer: Thank you for your suggestion. We have removed Fig. 9A, Fig 9B, and Fig. 9C (Figure 9 in previous version is now Figure 10), the graphs of unstained, single-stained propidium iodide (PI), and single-stained AV and have kept only DMSO control, and cATC (6e, 6j, 6N) treated group. All experiments have been repeated thrice and statistics have been performed. Kindly have a look at Figure. 10C, 10D, 10E, 10F, and 10G. Figure legend has been modified accordingly.

Comment 3: In Figure 10, the position of the DMSO group in the histogram is suggested to be placed to the left as in the DMSO group in the result of the wound healing assay.

Answer: Thank you for pointing it out, position of the DMSO group in histogram has been placed to left (Figure 10 in previous version is now Figure 11). Kindly have a look in Figure 11. 3

Comment 4: In Figure 11, the doses of cATC derivatives 6E, 6J, and 6N displayed on the resulting graph are inconsistent with the text, please correct this. Also, how are these doses screened out

Answer: Thank you for pointing it out, it was a typographical error from our side in the text. Figure 11 in previous version is now Figure 12. We have corrected the doses of cATC derivatives 6e, 6j and 6n in the text. Please see line 417-419

Comment 5: In addition to SNAIL and SLUG, are there other indicators representative of the EMT phenomenon that can simultaneously verify the inhibitory effect of cATC derivatives 6E, 6J, and 6N?

Answer: As suggested by you, we have detected two more EMT markers, E-Cadherin and NCadherin. Please refer the data in Figure 12 A, B, and C. Figure legend and the result section text (line 417-419) have also been modified.

Comment 6: In the discussion section, hinokitol and β -Thujaplicin should be the same intervention, why the effects be explained separately?

Answer: As suggested by you, we have combined the information regarding hinokitol and β Thujaplicin intervention in the discussion section. Please refer to the lines 467-469

Comment 7: According to the present data, please compare the efficacies of the three cATC derivatives (6e, 6j, and 6n) in the DNA binding affinities and anti-tumor activity.

Answer: As suggested by you, we have experimentally validated the efficacies of three cATC derivatives (6e, 6j, and 6n) in DNA binding by microscale thermophoresis (MST) technique and its affinity was evaluated by Hill's coefficient (1.6, 5.8, and 6.3 for 6e, 6j, 6n, respectively) as shown in Fig 7 E, F, and G. We have added this information in the result section (line 293-297). Lower IC50 values of 6j and 6n, compared to 6e (6e-25.42 μ M, 6j-19.23 μ M, and 6n-18.92 μ M) are as per Hill's coefficient. We have evaluated the anti-tumor activity by in vitro and in vivo experiments. Our observation shows decrease in cell proliferation, migration, tumor size as shown in Figure 8, Figure 11, and Figure 13, respectively. Whereas, our FACS data shows increase in apoptosis which is shown in Figure 9.

Comment 8: Authors showed that cATC derivatives (6e, 6j, and 6n) induced apoptotic cell death by FACS-based detection of FITC-labelled annexin V and propidium iodide staining. The population was not significantly increased in positive control PI group in figure 9B. On the contrary, an increased cell population stained with annexin V was observed in the positive control PI group. Authors should further confirm the data and repeat this experiment again.

Answer: Thank you for pointing it out, as per your suggestion we have repeated the experiment and we observed a significant increase in propidium iodide and annexin V. Please have a look in Supplementary information (SI) Figure S64.

Comment 9: In this study, authors conducted migration assays using the scratch assay and revealed that cATC derivatives reduced the migratory ability of HeLa cells after 24 h treatment. Simultaneously, cATC derivatives inhibited cell proliferation and induced apoptosis after 18 h treatment. What is the concentration of cATC derivatives displayed to 4 inhibit cell migration in figure 10? It is unable to rule out the possibility that decreased cell numbers in the area of the scratched wound are primarily due to cell death rather than inhibiting the migratory abilities. Accordingly, the inhibitory effects on cell migration may be overestimated. To perform the migration assay, authors have to consider the doubling time of cells and the effect on growth inhibition.

Answer: Thank you for bringing this aspect, in order to evaluate the migratory effect of HeLa we treated the cells with IC50 doses of cATC derivatives for 18 hours (doubling time of HeLa is 24 hours). The media was replaced with fresh media and images were captured at 24 h. We had to keep the treatment time 18h as nuclear localization was clearly visible at that time point. We agree that decrease in migration could be possibly due to cell death. We have used IC50 doses in migration assays to minimize the effect of cell death. Also, our Western blot in Figure 11 a-c show the increase in E-Cadherin and decrease in SNAIL, SLUG, and N-Cadherin expression at protein levels which are the major EMT markers and play an important role in migration.

Comment 10: Authors used different cancer cell types in vitro and in vivo. Please explain the reason.

Answer: We agree that we used different cell types for in vitro and in vivo experiments. So far, in publications use of Murine uterine cervix cancer (MUCC) and U27 cell lines has been reported, but

unfortunately, these cell lines are not available with ATCC or other commercial sources. As we wanted to evaluate these compounds' anti-cancer effects in vivo, we used the 4T1 cell line. We have added this point in the discussion. Please see 347-348.

Comment 11: Figure 2B is of poor quality. Please provide a replacement with high resolution. In the text, the authors describe that UV-Vis spectra and fluorescence spectra of 6a are shown in figure 5, while 6j was displayed in the legend. Similar errors are found. There is no description of the results shown in figure 7D to 7H. In the figure legend of figure 7, it is stated that “d-f, The cell cycle was evaluated by flow cytometry after staining with propidium iodide (PI)...” However, it seems that the results are cell viability and scavenging. The results of cell cycle are described in other paragraphs which are shown to display in Figure 9 a-h. Again, the results displayed in Figure 9 are actually the examination of apoptosis. Please check.

Answer: Thank you for indicating it, there was a typographical error. We have replaced figure 2B with better resolution. We have also corrected legend of Figure 5. Thank you for indicating it, there was a typographical error. During manuscript writing we did some reshuffling in figures, original Figure 7 d-f were moved to Figure 10 as A and B. In the revised manuscript we have corrected the description regarding Figure 8 (previously Figure 7) and Figure 10. Please see the figure legends for Figure 8 and 10. For the text description of Figure 8 D-F please see line 314-320.

Comment 12: Please show the error bars (standard deviation) and statistical difference if there are three independent tests.

Answer: Thank you for pointing it out, as per your suggestion we have shown error bars (standard deviation) calculated by three independent tests in Figure 8 A-F and Figure 10 G

Reviewer #3 (Remarks to the Author):

The manuscript having good novelty and highly significant work for the community with great application as anticancer activity. I recommend this manuscript after the correction of some flaws in the manuscript like

Comment 1. In scheme 1, give the exact amount of imine should be added as place of "excess" for troponyl dihydropyrazine.

Answer: We have used four equivalent of imine. The synthetic procedure is provided in the Supplemental Material.

Comment 2. In scheme 1, what is the difference between the structure of 6c and 6d, as it can be seen both the structure has been same.

Answer: Compound 6c and 6d are structurally different. Compound 6c is Phenethyl amine while 6d is benzylamine

Comment 3. It should be more appropriate if the author should have discussed the mechanism with FRET or PET way.

Answer: We have performed comparative studies of cATC product formation by recording the time-dependent fluorescence spectra of the reaction mixture that clearly show an enhancement of fluorescence intensity with respect to time. Thus the formation of new fluorescence product. Next our time-dependent mass analyses support the formation of cATC fluorescent products. Their spectra are provided in the Supplemental Material.

Comment 4. Also given the FTIR spectra in the Supplemental Material.

Answer: We have provided FTIR of all cATC derivatives. Their spectra are provided in the Supplemental Material.

Comment 5. Add some more references like doi.org/10.1021/ja3103007, doi.org/10.1002/jhet.4357 etc.

Answer: We have incorporated both references in the revised manuscript. Ref 34 and 35.

REVIEWERS' COMMENTS:

Reviewer #1 (Remarks to the Author):

The authors have addressed the issues raised in my previous review. Only a minor question persists: all the X-ray structures (cif files, Fig. 3 is not clear in this respect) and the DFT calculations indicate that the configuration of compounds cATC (6) is trans, with both carboxylate and aryl groups in axial disposition. However, the representations gathered in Figs. 2, 4, 5 and 6, as well as the structures reported in the Supplementary Material, are ambiguous or suggest a cis configuration. These drawings should be corrected.

Reviewer #2 (Remarks to the Author):

The author response to my comments well.
I think this manuscript can be accepted in the current form.

Reviewer #3 (Remarks to the Author):

The authors have modified/implemented the manuscript following the indications of the reviewers, therefore it can be accepted for publication.

Rebuttal Letter (Answers to Reviewers' Comments)

Reviewers' comments:

Reviewer #1 (Remarks to the Author):

The authors have addressed the issues raised in my previous review. Only a minor question persists: all the X-ray structures (cif files, Fig. 3 is not clear in this respect) and the DFT calculations indicate that the configuration of compounds cATC (6) is trans, with both carboxylate and aryl groups in axial disposition. However, the representations gathered in Figs. 2, 4, 5 and 6, as well as the structures reported in the Supplementary Material, are ambiguous or suggest a cis configuration. These drawings should be corrected.

Answer: Thank you for giving opportunity to revise manuscript. To achieve maximum 10 figures in the main manuscript, we have shifted three figures (Fig 3/Fig 5/Fig 6) from the main manuscript to the Supplementary material. Their ORTEP diagram, with other interactions, are also placed in Supplemental Material (Fig S1-S4). Their major products are trans. Now we have corrected in manuscript and Supplemental material/ Supplementary Data 1 files